# Efficient Secure Routing Mechanisms for the Low-Powered IoT Network: A Literature Review

**Muhammad Zunnurain Hussain \*** and **Zurina Mohd Hanapi**

Department of Communication Technology and Networking, Universiti Putra Malaysia,
Seri Kembangan 43400, Selangor, Malaysia
\* Correspondence: gs58270@student.upm.edu.my

**Abstract:** The Wireless Sensor Network in the Internet of Things (WSN-IoT) has been flourishing as another global breakthrough over the past few years. The WSN-IoT is reforming the way we live today by spreading through all areas of life, including the dangerous demographic aging crisis and the subsequent decline of jobs. For a company to increase revenues and cost-effectiveness growth should be customer-centered and agile within an organization. WSN-IoT networks have simultaneously faced threats, such as sniffing, spoofing, and intruders. However, WSN-IoT networks are often made up of multiple embedded devices (sensors and actuators) with limited resources that are joined via various connections in a low-power and lossy manner. However, to our knowledge, no research has yet been conducted into the security methods. Recently, a Contiki operating system's partial implementation of Routing Protocol for Low Power & Lossy Network RPL's security mechanisms was published, allowing us to evaluate RPL's security methods. This paper presents a critical analysis of security issues in the WSN-IoT and applications of WSN-IoT, along with network management details using machine learning. The paper gives insights into the Internet of Things in Low Power Networks (IoT-LPN) architecture, research challenges of the Internet of Things in Low Power Networks, network attacks in WSN-IoT infrastructures, and the significant WSN-IoT objectives that need to be accompanied by current WSN-IoT frameworks. Several applied WSN-IoT security mechanisms and recent contributions have been considered, and their boundaries have been stated to be a significant research area in the future. Moreover, various low-powered IoT protocols have been further discussed and evaluated, along with their limitations. Finally, a comparative analysis is performed to assess the proposed work's performance. The study shows that the proposed work covers a wide range of factors, whereas the rest of the research in the literature is limited.

**Keywords:** internet of things; industrial internet of things (IIoT); low powered; computer networks; Contiki; IoT security; network management; machine learning

## 1. Introduction

IoT has been thriving as another global innovation in the last few years. It is expected that the world's fortunes will be changed by implementing IoT in various systems over the coming years. IoT will likely revolutionize the way we live today. The Internet of Things foundation was established to improve communication and data exchange between humans and devices for massive data transfer [1]. IoT's motivation involves the association of registering gadgets, mechanical and computerized objects, humans, and machines through applications utilizing the web interface and portable applications. The IoT climate can move information through an organization without expecting human-to-human or human-to-computer correspondence [2].

IoT is becoming a significant necessity for many industrial and communication technology applications. There has been an enormous increase in IoT implementation as it has been considered to have the massive number of 50 billion devices connected to the Internet by 2020 [3]. Furthermore, IoT applications designed to assist the disabled or elderly provide

ease and mobility at varying degrees of unconventionality for a reasonable price [4]. In addition, IoT adds to numerous fields, for example, agribusiness, climate, clinical areas, the educational sector, transportation, and finance. These innovations and upgrades improve our everyday lives [5].

Figure 1 shows the graph depicting the number of IoT devices in billions globally from 2018 to 2030 [6].

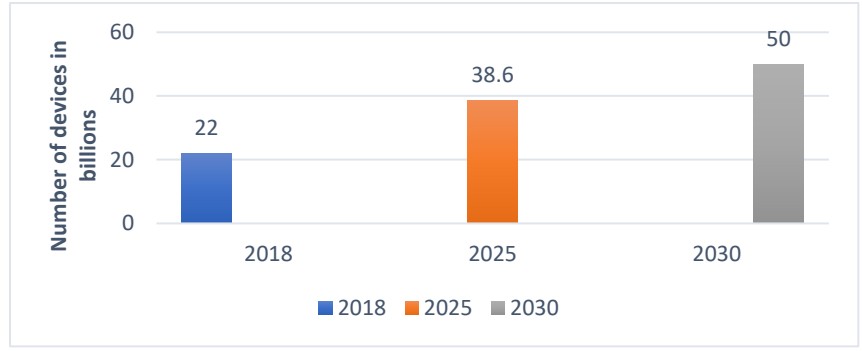

**Figure 1.** Number of IoT devices in billions connected globally from the year 2018 to 2030 [6].

Numerous organizations and scientific research associations are working on various aspects of the IoT. They have presented a functional outline for the IoT's impacts on the economy and the vast majority of other existing fields over the next 10 years. Cisco is the primary organization that delivers numerous IoT undertakings, which included 24 billion smart objects by 2019. It is also expected that the Huawei company will introduce 100 billion IoT associations by 2025 [7–9]. Every second in the world, 127 devices are linked to the Internet. By 2020, out of all electronic device use, 63% will be using IoT technology. Of all the massive, smart city commercial projects, 23% consist of IoT implementation, while by the end of 2020, 40% of all healthcare organizations were embedding IoT [10]. Figure 2 depicts an analysis of the worldwide expenditure in billions of dollars on IoT from 2018 to 2023, which shows the cost of IoT is increasing day by day as enhancement in technology (such as automated machine systems, devices, etc.,) increases. In 2018 the expenditure was 616 billion USD, after that it increased slightly in 2019 and there was a significant change predicted for 2022–2023 [11].

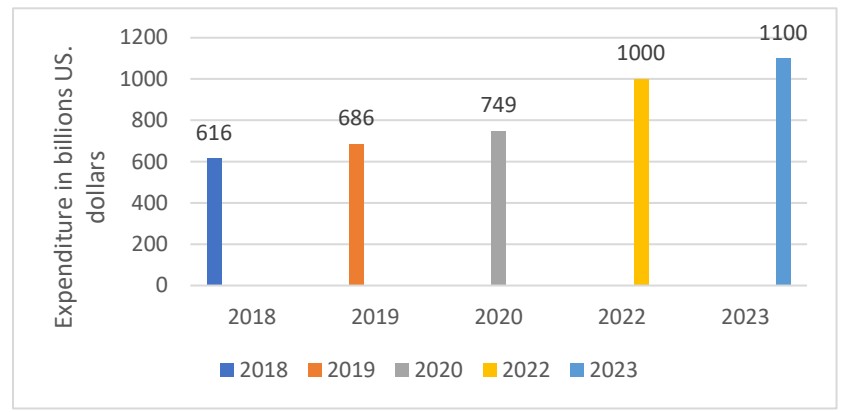

**Figure 2.** Worldwide expenditure on IoT from the year 2018–2023 [11].

Via controllers and cloud management, autonomy can be generated for the self-sufficiency and decision-making of nodes [12]. There is always a wide open door for intruders or hackers to utilize IoT devices for their potential benefit via various attacks, such as Denial-of-Service (DoS) attacks, phishing emails, and other unsafe worms or Trojans [13]. The IoT layers present multiple risks such as sniffing, spoofing, eavesdropping, and intrusion. IoT utilizes hubs, sensors, and intelligent recognition gadgets to gather

information. Because of the nonappearance of verification administration, unapproved access can change data integrity or even erase the stored data [14,15].

IoT systems can work under various conditions and, in most cases, have little computing capacity. Therefore, specific IoT devices can connect to many hubs, raising significant security concerns. As a result, security issues have proven to be more challenging to solve, as it is difficult to establish a nonexclusive security architecture or model [16]. The Internet has undergone remarkable changes that offer both extraordinary opportunities and significant difficulties for users; troubles emerge from unauthorized users utilizing cyberspace and exploiting its numerous weaknesses. Various cyber insights are required for the Internet to assess risks and overcome challenges [17].

The increasing proliferation of WSN devices in an actuating–communicating network has spawned the Internet of Things (IoT), in which data is seamlessly shared across platforms by fusing sensors and actuators with our surroundings. Medical and environmental monitoring can be automated using these low-cost WSN devices. RPL improves the utilization of these sensors in real-world applications by assessing their performance. Low-Power and Lossy Networks (LLNs) are mainly restricted nodes with limited processing power and fluctuating energy. Most traffic patterns are multipoint-to-point or multipoint-to-multipoint rather than point-to-point. As a result, data rates are often reduced, resulting in instability [18]. Contiki is an operating system that allows RPL and lossless monitoring of Internet of Things devices. Topological node assignment is based on multi-hop transmissions and has been employed in environmental monitoring, health care, and other smart systems [19]. Routing is a popular topic in the IoT community because of the limitations imposed by these devices. In many IoT networks, the Internet Engineering Task Force's (IETF) routing protocol for low power and lossy networks (LPN) has become the norm since it was intended to effectively utilize the finite resources of IoT devices while delivering effective routing services. RPL's architecture included many but optional security methods for ensuring reliable routing. Research on the security elements of RPL's routing protocol, such as routing assaults, novel mitigation mechanisms and intrusion detection systems (IDSs), and goal functions with an eye on security, has exploded since the protocol's 2012 standardization (OFs). The impacts of RPL's security features against routing assaults have not yet been studied, which is strange. RPL's security features have not been implemented in any of the existing IoT operating systems (OSs), such as Contiki OS and TinyOS.

RPL security features have been partially implemented in Contiki OS, with the addition of a preloaded secure mode (PSM) and an optional replay protection system. Using this approach, we were able to provide the groundwork for this paper. We summarize our contributions as follows.

We confirmed that, except for the wormhole, RPL in the preinstalled secure mode (PSM) could prevent external adversaries from entering the IoT network for the examined attacks (WH). Additionally, we demonstrated that the optional replay protection offers superior protection against the neighbor attack (NA). However, it needs more optimization to minimize its impact on energy usage. We observed and assessed the effect of the examined assaults on the routing topology and offered two simple strategies for mitigating the consequences of the investigated attacks without using external security measures, such as intrusion detection systems or other security mechanisms. Another performance comparison of the suggested methodologies' implementation was undertaken. The findings indicated that RPL performed better in terms of end-to-end (E2E) latency and packet delivery rate (PDR) when subjected to Selective-Forward (SF) and Black-hole assaults.

This paper explored the numerous security challenges in wireless sensor networks (WSNs) and the IoT. The function of the IoT in Industry 4.0 demonstrates how various automated systems can be used to optimize processes. Similarly, the IoT-LPN protocols are examined in this paper, which describes each protocol's strengths and weaknesses. This paper also discusses security objectives (availability, confidentiality, privacy, and so on), threats, and WSN and IoT-LPN problems. These attacks are further categorized into physical layer-based, network layer-based, software-based, and data-based attacks. In

addition, the security processes associated with Industry 4.0 are addressed. The suggested research performed a comparative analysis with current work based on various criteria, indicating that the proposed work covers several factors, whereas existing work discusses fewer. Based on the preceding explanation, the proposed work focuses on four questions and targets them as follows:

Q1.  What are the applications utilized in IoT-LPN?
Q2.  What are the existing protocols in IoT-LPN and their strengths and limitations?
Q3.  What are the security objectives of WSN-IoT?
Q4.  What are the security issues and challenges in WSN-IoT?

Figure 3 depicts the research collection mechanism, including identifying the data, title screening, exclusion criteria, and finally, the included papers.

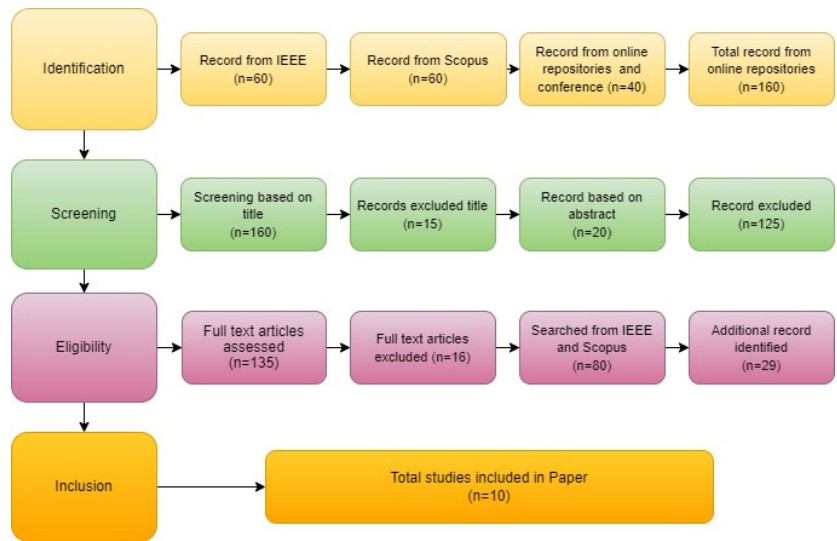

**Figure 3.** Research collection mechanism.

The rest of the paper can be described as follows. Section 2 briefly discusses IoT in Industry 4.0. Similarly, Section 3 is related to IoT-LPN architecture and its applications. Section 4 is about security issues and challenges in WSN-IoT. Section 5 covers the categories of network attacks in WSN-IoT layers. Section 6 is related to the WSN-IoT security mechanism in Industry 4.0. Subsequently, the existing literature is discussed in Section 7. Similarly, a comparative analysis of the proposed work with the current work is discussed in Section 8, and finally, the conclusion and findings are described in Section 9.

## 2. IoT in Industry 4.0

A company should adapt its manufacturing and logistics processes to follow emerging technology in this modern era. Profitability and cost-effectiveness have increased as business has expanded. Customer-centric development methods should be included, and internal business agility should be prioritized. Similarly, the transformation of all social and business structures around digital communication can be described as "digitalization." Digital innovations are incorporated into corporate activity by digitalizing all conceivable operations [20]. However, IoT provides the opportunity to view data from anywhere and share data between computers, devices, and nodes. Manufacturing processes are interconnected and there are real-time flows between all aspects of the supply chain [21]. The IIoT is a particular IoT field that emphasizes its implementations and uses for new industries and intellectual development. It is a dynamic structure with a wide range of processes. Besides, it is a central feature of the manufacturing sphere and is closely tied to the fourth industrial revolution (IR 4.0) [22]. It combines several cutting-edge critical technologies to produce a system that outperforms the sum of its parts. This one-of-a-kind domain stands out for its many innovative applications and services and its myriad integrated appliances and modern manufacturing operations [23]. Through the use of

core technologies such as Cyber-Physical Systems (CPSs); IoT; autonomous, scalable, cooperative robotics; the Internet of Services (IoS); simulations that exploit real-time data to create a computer model that reflects the actual world; big data analytics; and enhanced re-assembly, IR 4.0 aims to improve and update existing production plants, maintenance and management processes, and technology to an intellectual level [24]. The term "responsibility" refers to determining whether a person is responsible for their actions. A top-down and bottom-up system would create an integrated network that helps an automated supply chain build skills, functions, divisions, and companies. The presentation, analysis, and interpretation of data from various sources, including industrial processes, warehouses, and corporate consumer information systems, has become the norm in IR 4.0 to aid real-time decision-making. Networking technology, smartphones, sensors, applications, middleware, and storage devices are all included.

Simply put, Industry 4.0 alters both living and working practices. However, a new era in human history becomes possible through technological developments comparable to the first, second, and third industrial revolutions. They provide more automation and act as a bridge between the physical and digital worlds by using a cyber-physical system (CPS), which is a fundamental component of the smart factory as envisioned by Industry 4.0. CPS is a manufacturing system that uses sensors and software throughout the manufacturing process. Sensors collect and preserve data, which is subsequently analyzed by a computer to make various decisions. These decisions immediately impact the physical system via actuators and human-machine interfaces.

Furthermore, CPS enhances automated machines with the help of industrial IoT. CPS also collects and distributes data from and about the plant's assets and locations. Therefore, various approaches (cloud computing, AI, and machine learning) have been utilized to analyze this data and make decisions that improve system optimization. CPS and IoT work in tandem to develop smart factories. These competitive factories have decreased downtime, increased efficiency, produced better products, and increased output [21]. A modified version of AntHocNet offers a unique routing mechanism for FANET. Compared to other traditional optimal path selection strategies, ant colony optimization, or metaheuristics in general, it has proven more reliable and effective. This study's energy stabilizing parameter enhances network performance overall and energy efficiency. According to the simulation findings, the suggested protocol outperforms generic Ant Colony Optimization (ACO) and other established routing protocols used in FANET [25].

## 3. IoT-LPN Architecture and Its Applications

The Internet of Things employs low-power and lossy networks, known as Low-Power and Lossy Networks (LLN), which may impose limits on infrastructure integration. It enables devices to interact with embedded devices, such as sensors, and can connect many nodes. The traffic variety of LLN systems is also defined; they use point-to-point, point-to-multipoint, and multipoint-to-multipoint architectures. Because of the intricacy of such a network, it is critical to have a routing protocol that serves the purpose. This has been one of the researchers' primary problems. So, to achieve this goal, the IETF ROLL working group developed RPL, a protocol for LLNs. This protocol is built on a collection-based network in which nodes gather information at regular intervals and transfer it to the collection point. The entire communication architecture is built on low-power wide area networks (LPWA) using unlicensed spectrum (Sigfox, LoRa) and other LPWA technologies proposed by the 3rd Generation Partnership Project (3GPP) that works within a licensed frequency range (NB-IoT, LTE-M). At the same time, the unlicensed spectrum origins made it more challenging to meet the integration goal and increased the possibility of interference and congestion. A licensed frequency range reduces external interference and improves dependability, signal-to-interference-plus-noise ratio (SINR), and security. Similarly, getting a license for these bands comes with a high upfront cost and a regular renewal price. The rise in cost will inevitably be passed on to subscribers, raising capital expenditures for deployment and ongoing operational expenses.

Software-defined networking (SDN) architectural technology increases network performance and monitoring [26,27]. However, the network system is divided into device

management, the Internet of Everything (IoE) gateway, and intelligent LPWA with the help of AI and deep learning. IoE services provide cellular communication in the licensed and unlicensed spectrum. Similarly, AI is responsible for smart wireless communication technology using smart applications and IoE services. Some typical IoT applications developed with the help of LPWA are the smart city, track and trace, and smart building applications.

## 4. IoT-LPN Protocols

Several routing protocols have been developed to improve the efficiency and functionality of networks in IoT systems. Low-powered protocols have been prevalent in the demand for low-powered IoT frameworks as they are efficient and require fewer resources, making them practical and providing many benefits. Table 1 shows the most efficient and popular low-powered and LLN protocols; here, the protocols are characterized by their foundational, low-powered protocols, including RPL, GOAFR, LOADng, SMRF Smart-Hop, and SPEED. GeoRank aims to improve P2P functionality and minimize the number of control messages needed, but it reduces scalability and requires static nodes or GPS-enabled devices. Further, the protocols are mapped with the routing solutions they present, which are P2P support, multicast communication, mobile node support, Quality of Service QoS, and energy efficiency. Table 1 briefly describes each protocol and states its strengths and limitations. The limitations of each protocol highlight grey areas that need attention for improvements. Energy-efficient region-based RPL (ER-RPL) is designed to prevent the network from flooding with peer-to-peer (P2P) route-finding packets, resulting in energy savings and an increase in the P2P packet delivery ratio. P2P-RPL allows for the construction of alternative P2P routes for application routing needs, but it increases the overheads and energy consumption of the network. Bidirectional multicast RPL forwarding (BMRF) improves both upstream and downstream multicast data forwarding. Still, it has a slight increase in memory consumption and can have low productivity due to end-to-end latency and incorrect parameter settings. Stateless Multicast RPL Forwarding SMRF improves RPL's multicast data forwarding and reduces energy waste but only allows for downward multicast broadcasting and can have high end-to-end latency. mRPL provides quick and reliable mobility support in RPL but increases the length of control messages and the number of control messages sent and received. Backpressure RPL (BRPL) aims to improve RPL's performance in large-scale networks, but it requires a large amount of memory and has a high end-to-end latency.

**Table 1.** Effective low-powered and LLN protocols in IoT frameworks.

| Protocol | Foundation | Routing Solution | Description | Strengths | Limitations |
|---|---|---|---|---|---|
| ER-RPL [28] | RPL | P2P support | ER-RPL prevents the whole network from flooding with P2P route-finding packets, lowering network energy usage. Furthermore, the technique enables the transmission of P2P messages and utilizes the structure produced by default RPL for various traffic patterns. | It prevents the whole network from flooding with P2P control messages, resulting in significant energy savings and an increase in the P2P packet delivery ratio. | Some location-aware nodes are required (e.g., GPS) to have a complex strategy; additional control messages are added in addition to the essential RPL messages. |
| P2P-RPL [29,30] | RPL | P2P support | P2P-RPL allows RPL networks to have better P2P data traffic and develops new P2P routes on demand as an alternative to RPL-built P2P routes. Communications sent between these nodes follow a single path. | Allows for constructing alternative P2P routes to meet application routing needs while avoiding using a root node for P2P message forwarding. | Increases the overheads and energy consumption of the network by flooding it with control packets. |

**Table 1.** *Cont.*

| Protocol | Foundation | Routing Solution | Description | Strengths | Limitations |
|---|---|---|---|---|---|
| geographic routing approachGeo-Rank [31] | RLP and Greedy Other Adaptive Face Routing GOAFR | P2P support | GeoRank seeks to improve 6LoWPAN's P2P functionality and minimize the number of control messages needed. It first uses the list of DODAG roots to determine the distance between the source and destination. The node transmits a message to its neighbor with one hop to the message destination and then passes the packet to the referenced parent. | Reduces scalability by avoiding the usage of DAO messages, allowing memory consumption. | RPL control messages are changed by default. Static nodes or GPS-enabled devices are required. |
| BMRF [32] | SMRF and RPL | Multicast communication | Except for the root node, when a node in the BMRF wants to send a multicast message, it sends it upstream and downward. Upward sending is conducted using unicast transmission, whereas downward sending is accomplished through the BMRF mode; this behavior aids in avoiding packet duplication. | SMRF improves both upstream and downstream multicast data forwarding. Increases the packet delivery ratio while reducing the number of radio transmissions and energy usage. | Increases memory consumption by a small amount. End-to-end latency and incorrect parameter settings can lead to low productivity. |
| SMRF [33] | RPL | Multicast communication | SMRF presents a cross-layer technique for multicast forwarding operations that enhances the working of Radio Duty Cycling (RDC) protocols. SMRF operates in storage mode with IETF multicast support. When compared to the default strategy, the conventional solution provides better outcomes. | RPL's multicast data forwarding should be improved. Reduces the energy waste caused by multicast RPL's many unicast transmissions. The processing of duplicated multicast packets is avoided. | Only downward multicast broadcasting is permitted. High end-to-end latency is possible. SMRF processes only packets transmitted from the node's selected parent. Messages received from child nodes are not processed or forwarded as a result. |
| mRPL [34,35] | RPL and Smart-HOP | Mobile node support | Smart-HOP employs RPL-control messages similar to beacons in mRPL. The mechanism is separated into two phases due to two types of nodes: mobile nodes (MN) and access points (AP). An MN broadcasts a sequence of n DIS beacons to a serving AP during the data transfer phase. Based on the n received messages, the serving AP computes the average received signal strength indication (RSSI) value. | In RPL, provides quick and dependable mobility support. Provides a technique for avoiding collisions and loops. Interoperable with RPL by default. Reduces packet loss and delay rate. | A small amount increases the length of the control messages. Increases the number of control messages sent and received. |

**Table 1.** *Cont.*

| Protocol | Foundation | Routing Solution | Description | Strengths | Limitations |
|---|---|---|---|---|---|
| BRPL [36] | RPL and backpressure routing | Mobile node support | BRPL supports the creation of numerous logical topologies based on distinct Optical Fiber(OFs). For each Directed Acyclic Grap (DAG), each Backpressure RPL BRPL node maintains a buffered packet queue. DODAG Information Object (DIO) messages communicate information between nodes, such as the maximum queue length, RPL rank, and queue length. As a result, each node that receives a DIO message must update the information about the packet sender in the neighbor database. | Improves RPL to accommodate mobility and dynamic traffic loads, reduces packet loss significantly, and may exist together in a network using default RPL. | End-to-end latency is momentarily increased. |
| Emergency Response IoT based on Global Information Decision- ERGID [37] | SPEED | QoS and Energy Efficiency | ERGID is a routing system for IoT applications that promises rapid emergency response and reliable data delivery. It relies on two distinct methods. For example, the delay iterative method (DIM) classifies the nodes of a potential route based on a global delay calculation. This method is used to prevent a legitimate route from being ignored. The second method, residual energy probability choice (REPC), allows for residual energy information throughout the message-forwarding process. | For emergency response applications, it provides reduced latency and load balancing. Reduces average latency and packet loss rate without increasing energy usage by selecting routes based on global information. | Uses many control messages to keep delayed information current and accurate. Calculations and routing table updates are required regularly. |
| Quality of Services (QoS) RPL [38] | RPL | QoS and Energy Efficiency | QoS RPL is a routing metric based on transmission delay that aims to better meet the energy efficiency and QoS criteria in LLNs. During the routing protocol operation, the information regarding energy and latency is mapped onto the control messages, while each node that receives a packet computes and updates this information. | Improves LLN energy efficiency and QoS while lowering delay and energy usage. | Reduces the ratio of packets delivered, which can disrupt load balancing, leading to an increase in control message overhearing. |

**Table 1.** *Cont.*

| Protocol | Foundation | Routing Solution | Description | Strengths | Limitations |
|---|---|---|---|---|---|
| LOAGng-IoT [39] | LOADng | QoS and Energy Efficiency | LOADng-IoT also has a route caching technique to decrease route search costs and a new error message for when an IC node loses its Internet connection. As a result, the idea enables LOADng to address better the QoS and energy efficiency needs of various IoT applications. | Allows LOADng to identify and maintain routes in diverse networks more efficiently. Allows nodes to identify Internet gateways without prior setup and provides a strategy for reducing the overhead created during route discovery. | Also necessitates the addition of an extra field to the average LOADng control messages. Memory use may be increased by using the route cache method. |
| RPLca+ [40] | RPL | QoS and Energy Efficiency | RPLca+ comprises two specialized libraries, one for estimating link quality and the other for managing neighbor tables. The first library aims to improve RPL operations using a hybrid link-monitoring architecture to evaluate connection quality with minimal overheads. The second library consists of a series of approaches for neighbor table management. | Improves data delivery reliability in RPL and offers a dynamic link; the quality estimator establishes the policies for the project. Routing table management enhances packet delivery rates. | Boosts energy usage by submitting implementation overheads. |

## 5. IoT-LPN Research Challenges

So, here's a quick rundown of the IoT-LLN challenges. We have seen that the entire communication strategy is based on LPWANs in an IoT-LLN. For LPWANs, scalability is a big challenge in the dense network [41]. It enables several devices to connect to each base station and deploy more stations across the network. Resultantly, structural scalability was already insufficient to meet LPWANs' use cases, so it required more devices to meet the requirements.

Similarly, most LPWANs are confined to star topologies. In contrast, cellular-based networks (EC-GSM-IoT, NB-IoT, LTE Cat. M1, 5G) depend on wired infrastructure to integrate networks and cover wider regions. So, the improper infrastructure hampers applications such as the agriculture IoT [42]. The scalability of short-range and cellular wireless networks is the subject of current research. Offloading (from the licensed to the unlicensed spectrum), common in cellular-based technologies, is impractical for LPWANs operating in the unlicensed spectrum. To overcome the scalability issues, there is a need to approach some other strategies, such as adaptive data rate MAC protocols, the adaptation of spectrum-efficient modulation techniques, and LPWAN channel diversity exploration. Another significant issue is the collection of LPWAN-relevant data regarding methodologies and performances. Because the data of popular LPWANs (LoRaWAN, SigFox, and NB-IoT) is easily accessible, gathering the data for others is complicated due to fewer references. Nowadays, LPWANs are widespread and there is more demand among users to develop new applications because of the discovery of new methods applicable to their personal lives and business operations.

It is understood that security and privacy are the primary concerns in all fields. However, there has been little emphasis on LPWAN's security in general. Unauthorized access can easily breach the security of a smart home controller. Using unauthorized

access, attackers can steal information and completely control home appliances, causing inconvenience to their users.

Similarly, unauthorized access to smart cities, agriculture, and inter-vehicle communication can cause death and environmental harm. So there is a need for adequate security to authenticate the user or owner efficiently; otherwise, LPWANs are not viable for commercial purposes [43]. Moreover, the essential components of security related to WSN-IoT are discussed in Section 4. These components are considered necessary before implementing any WSN-IoT application; otherwise, it will be vulnerable.

## 6. Security Objectives of WSN-IOT

WSN-IoT's security requirements are the essential characteristics necessary to be implemented to fulfill network security requirements. It consists of various preventive measures for the smooth functioning of the IoT framework [4,44–49].

### 6.1. Availability

The nature of keeping the service accessible to clients is accessibility. The goal of accessibility is to provide clients with the ability to obtain services at any time and from any location. It is critical to keep assets regularly available to clients and the organization. Consequently, all clients must be confirmed to combat assaults and risks to the organization. Accessibility may help to avoid blockage circumstances such as framework conflicts and organizational blockages that disrupt the information flow.

### 6.2. Accountability

Accountability is one of the WSN-IoT framework's basic properties, but it cannot preempt network attack risks and WSN-IoT vulnerabilities. However, rationing and supporting other security criteria such as data integrity and privacy are imperative. They are utilized to follow any node (device) that sends and receives information to notice and distinguish any obscure activities by providing guidelines for the device, clients, and their actions.

### 6.3. Confidentiality and Privacy

Confidentiality is otherwise known as privacy To fulfill the security requirements, it is implemented to prevent unauthorized clients from obtaining information. Confidentiality gives recognizable proof of verification and authorization for any sensitive item in the IoT network. Numerous security modules ensure the security of information. Maintaining data secrecy is a critical security requirement as it is vital to keep the framework intruder-proof. Privacy guarantees authorized users' private data and preempts intruders from accessing network services or stealing any data. Privacy has to be implemented at many levels. Privacy for devices is necessary to maintain physical and data confidentiality, as a network can be exposed to data intrusion. Privacy during data transmission within IoT devices preserves sensitive information. Privacy is crucial during the processing and storing of data, as it is most vulnerable at this point. Privacy of location is intended to prevent the disclosure of the geographical position of IoT devices from intruders.

### 6.4. Auditing

Auditing is essential; without it, the framework's criteria for meeting security requirements will not be accomplished. It is used to recognize the security shortcomings of WSN-IoT. Auditing is entirely related to accountability, yet it depends on assessing the framework and its services. Auditing measures how well the WSN-IoT framework meets its network performance criteria and components.

### 6.5. Integrity

Integrity is one security idea that empowers legitimate and authorized access to modify data according to requirements under limited conditions. Integrity can forestall

inner attacks, the most hazardous issue in the network framework, as all users must be validated and authorized with access rights. Notwithstanding, cybercriminals may change information during network communication. Integrity may preempt outside attacks to get to or alter sensitive information.

### 6.6. Access Control

Network access control is verified by an authorized network administrator for the smooth management of user access. It gives clients/users explicit roles or verified admittance to utilize network assets to view, alter, or modify data. Access control offers certain rights to legitimate users to perform precise work.

### 6.7. Authentication and Authorization

Authentication is the user's verification, the primary security necessity, as it recognizes users as validated clients utilizing security frameworks such as cryptography algorithms. After authentication, authorization plays a role in the approval of authentic users to use network services. Table 2 shows the security objectives of WSN-IoT.

**Table 2.** Security objectives of WSN-IoT.

| Sr. No | Security Objectives of WSN-IoT | Definition | Layers |
|---|---|---|---|
| 1 | Availability | Accessibility is the nature of keeping the service accessible for clients. | Perception Layer |
| 2 | Accountability | Accountability is one of the IoT frameworks' basic properties, but it cannot preempt network attacks and IoT vulnerabilities. | Network Layer |
| 3 | Confidentiality and Privacy | Confidentiality is otherwise known as as secrecy; to fulfill the security requirements, it is implemented to prevent unauthorized clients from getting information. | Network Layer |
| 4 | Auditing | This is an essential component; without it, the framework's criteria for meeting security requirements will not be accomplished. It is used to recognize the security shortcomings of the IoT. | Network Layer |
| 5 | Integrity | It enables legitimate and authorized access to modify data following requirements, but only under certain conditions. | Network Layer |
| 6 | Access Control | It gives clients/users explicit roles or verified admittance to utilize network assets to view, alter, or modify data. | Application Layer |
| 7 | Authentication and Authorization | Authentication is the user's verification, which is the primary security necessity as it recognizes users as validated clients utilizing security frameworks such as cryptography algorithms. | Network, Application Layer |

## 7. Security Issues and Challenges in WSN-IOT

### 7.1. Data Confidentiality

In the field of WSN-IoT and network protection, data secrecy is a critical concern. The client has access to the details and the system management in WSN-IoT frameworks. The IoT device should check that the user or machine has been granted access to the system [50]. Approval determines whether a person or device can receive assistance after presenting distinguishing evidence. Access management restricts property access by granting or refusing permission based on a series of laws. Creating a secure connection between devices and services necessitates approval and access control. The main point is creating a specific relationship between other devices and administrations, which requires

support and access control. The most critical problem in this situation is making access management regulations easy to develop and understand. This is a vital issue in the Internet of Things; many clients, objects, and devices must verify each other through trustworthy administrations to gain system access. The problem is to find a solution for safely dealing with the client's personality, items, and gadgets [51,52].

### 7.2. Privacy

Privacy and confidentiality are significant issues in WSN-IoT gadgets and frameworks under the IoT systems' universal character. Entities are linked, and information is conveyed and exchanged via the Internet, delivering client protection and causing various risks to sensitive information in many ways. So that the exploration issues are satisfied, knowledge acquisition security is just as important as information sharing security. Information protection is one of the primary uncertainties in the WSN-IoT because of the high chance of security vulnerabilities, such as sniffing and spoofing, unapproved access, data altering, and forgery with the unapproved altering of IoT nodes [53]. An aggressor can exploit numerous WSN-IoT administrations and applications to store sensitive and personal data, and if they are exposed, unstable and sensitive data can be exposed to outsiders [50,54].

### 7.3. Trust Management

In WSN-IoT frameworks, there is a consequence of regional conventions, resources, and limits of distinctive devices, which is a considerable assessment of IoT trust management. Trust is a significant part of WSN-IoT security, data security, administration, applications, and client protection. Trust is a fundamental component of communications among WSN-IoT devices to trade and manage information. IoT layers have a unique assortment of gadgets. Every gadget creates an enormous amount of information vulnerable to various assaults, dangers, and issues. These issues and attacks have the potential to spread across all IoT layers. As a result, the accuracy of information and administration will be reduced [53,55].

Trust management in IoT ought to accomplish the accompanying objectives of having faith in IoT nodes and choices to help one another. It should moderate client security, information transmission, and trust correspondence, as indicated by the IoT system's strategy. It should increase the superiority of IoT services, framework security, and reliability [56–58]. Furthermore, clients should not be aware of it.

### 7.4. Vulnerabilities

Vulnerabilities are flaws, and flaws in a system or plan that allow attackers to run commands, access unapproved data, and trigger DoS. In WSN-IoT implementations, bugs may be identified in several locations. They can be weaknesses in the client's devices and flaws in the system's hardware, code, or techniques used in the methods [59]. Hardware and software are the two fundamental components of IoT architecture. Both have configuration flaws daily. Hardware loopholes are challenging to detect and repair, regardless of whether the vulnerabilities were identified due to equipment similarities and interoperability, or the effort required to overcome them [16]. They can be found in working systems, application programming, and control programs, such as communications conventions and software changes. A significant cause of exposure is human error. The consequences of not understanding the necessities of teamwork, requirement engineering, testing and validation, security assessment, data integrity, and privacy can cause the framework to fail [60].

### 7.5. Security

Physical, network, and data protection are significant issues in WS-IoT frameworks. The growth in the number of connected devices on communication networks in the IoT [61] leads to increased security risks and new security challenges. Protection risks are acquired by any node that connects to the Internet, whether it is a limited or smart device [62]. On the Internet of Things, you can find almost any security issue. As a result, a few primary

security criteria in the IoT, such as acceptance, confirmation, classification, confidence, and information security, should be considered.

Consequently, things should be safely associated with their assigned networks, flexibly controlled, authenticated, and authorized [63]. Physical security tampering, stealing, and attacks are performed on IoT devices. The attacker can grab and steal a node or exchange it with a malicious node, causing harm to the whole network; moreover, the intruder can break the node or steal valuable or peculiar information that could be used against the system [64]. Maintaining a secure network means stopping intruders from finding their way into the system and causing severe damage by sending malware, sniffing, spoofing, stealing sensitive data, man-in-the-middle eavesdropping, or performing DoS attacks. Outsiders or employees within the organization can be intruders. Data security entails ensuring data integrity and privacy while data is transferred within the framework. Security is a method of protecting information from tyrannical forces or unauthorized access. IoT security depends heavily on information security, also known as computer security. [65].

### 7.6. Interoperability

A fractured landscape hampers users' value with patented IoT technical execution. Even if complete interoperability across goods and services is not always possible, consumers cannot like purchasing products and services that lack versatility and are subject to distributor lock-in [66]. Poorly designed WSN-IoT devices can negatively impact the networking resources to which they are linked. Another significant factor is cryptography, which has been used for years to protect against security vulnerabilities in several applications. A single protection application cannot have a suitable defense mechanism against attacks [67]. As a result, various levels of security are required to counteract WSN-IoT authentication risks. Hacks could be avoided by designing more sophisticated security features and incorporating them into devices. This evasion occurs because consumers purchase goods with good security features to guard against vulnerabilities. Any of the steps suggested to guarantee that the IoT is safe are cyber-security mechanisms [68].

### 7.7. Identification, Authentication, and Authorization

Nodes are the IoT building blocks that need to be defined in the network or physically. IoT networks cover a large area to track the transmission between devices and acquire access to the entire network. The total naming layout of nodes is unsafe without data consistency [69]. DNS cache positioning assaults may wreak havoc on the network's overall performance. For each target to be uniquely identified, node identification is necessary. The false node should be detected efficiently since each mark indicates a potential attack location. The network must be defended against physical or logical attacks on devices and their data. Authentication requires checking the identity of the nodes [70]. Undeniably, if contact with the correct node is not ensured, the secrecy and fairness of the messages exchanged cannot be guaranteed. An attacker can access the network and insert erroneous statements if the authentication is poorly handled. It is challenging to ensure authentication because of the wireless media's existence and the nature of sensor networks. Authentication involves confirming that you are who you claim to be. This is commonly achieved using an authentication method based on a username and password [71]. This scheme, though, is not safe enough. Passwords typically need to be updated regularly, and unattended computers should not be used. Authentication also requires the authentication method for both the sender and the recipient to validate the messages' origin [72].

### 7.8. Attacks

"The IoT frameworks hold a vast volume of information; the network layer is particularly vulnerable to attacks, creating much network congestion." The network's data integrity and authentication are critical security problems [73]. A significant problem is an attack by hackers and rogue nodes that damage the network's computers. The current security restrictions applied to IoT render them susceptible to attacks. Based on the particu-

lar design and features of the WSNs, these attacks usually follow new tactics [74]. Indeed, in the Open System Interconnection (OSI) model, attacks can be characterized according to the targeted protocol layer. Another method of grouping classifies assaults depending on the existence of the offender. We may also describe attacks as internal or external, passive or active. Nodes outside the network execute external attacks [75].

Passive threats are confined exclusively to the study, capture, and data snooping of traffic. Active attacks, however, usually exploit the data by disrupting the connection between the nodes and affecting the nodes' availability, so attacks can also be carried out [76]. On the other hand, internal attacks are initiated by valid network nodes that function against their requirements. Table 3 shows the security issues and challenges in IoT.

**Table 3.** Security challenges in WSN-IoT.

| Sr. No | Services | Challenges |
|---|---|---|
| 1 | Data Confidentiality | Data confidentiality is a significant concern in WSN-IoT frameworks since the customer can access the details and system administration. As a result, an IoT device access management system is required to identify the legitimate user and allow system access. |
| 2 | Privacy | Entities are linked, and information is conveyed and traded across the Internet, providing client safety while posing various hazards to sensitive data in numerous ways. There is a high chance of security vulnerabilities, such as sniffing and spoofing, unapproved access, and data alteration through unauthorized modification of WSN-IoT nodes. |
| 3 | Trust management | IoT layers have a distinct and diverse set of devices. Every device generates massive amounts of data subject to numerous attacks, risks, and concerns. These flaws are extended to each IoT layer and highly affect the quality of information or administration. |
| 4 | Vulnerabilities | Vulnerabilities are the loopholes that allow the attacker to access the device unethically and steal important information. Vulnerabilities in WSN-IoT exist at numerous levels, including user devices, scripts, hardware, methodologies, etc., which cause the framework to fail. |
| 5 | Security | Security is a significant concern in all fields. The growth in the number of connected devices to communication networks in the Internet of Things [61] leads to increased security risks and new security challenges, such as unauthorized access. With the help of unauthorized access, attackers can steal information and take complete control of smart appliances which causes inconvenience to their users. |
| 6 | Interoperability | WSN-IoT devices that are poorly designed might harm network resources. However, the customer rarely pays attention to the products and services that lack versatility. |
| 7 | Authentication, Identification, and Authorization | An attacker can access the network by inserting erroneous statements if the authentication is poorly handled. However, it is challenging to ensure authentication nowadays because of the wireless media's existence and the nature of sensor networks. |
| 8 | Attacks | An important issue is an assault by hackers and rogue nodes that damages the machines on the network. The current security limits imposed on WSN-IoT render it vulnerable to various threats. |

## 8. Categories of Network Attacks in WSN-IoT Layers

Network attacks can be classified based on the specific IoT layer they hit. However the categories of network attacks on IoT layers are Physical. Network, Software and Data Attacks [77].

*8.1. Physical Layer Attacks*

(1)    Tampering

A denial of service (DoS) attack occurs when network tampering detaches or changes the existing network. The attacker can replace the actual node with a dummy or malicious node. After demonstrating the physical theft of devices, the attacker may extract sensitive data from the captured devices to launch other attacks in the WSN-IoT framework [78].

(2)    Malicious code injection attack

Code injection or remote code execution occurs when an attacker utilizes a software input validation error to construct and execute malicious code, known as code injection or remote code execution [79]. The server-side interpreter injects and executes code into the language of the targeted application. Any software that takes invalidated input directly is vulnerable to code injection, and online apps are a common target for hackers [80].

(3)    RF interference jamming

To perform DoS attacks on Radio Frequency Identification (RFID) tags/sensor nodes, an attacker generates and delivers noise signals over radio frequency [17]/WSN signals [81]. All devices communicate via radio and operate on a wireless signal [17]. If a more powerful signal overshadows such signals, a "stronger" signal that drowns out the standard wireless frequency is jamming.

(4)    Fake node injection attack

Many infected nodes collaborate to create a bogus report and insert it into the network in false node injection attacks. An attacker places a faulty node between them to monitor the data transfer between two valid network nodes. Since they only validate a fixed number of message authentication codes (MACs) held in the files, these attacks are brutal to defend against [82].

(5)    Sleep denial attack

Denial-of-sleep (DoSL) is a denial-of-service attack that prevents sensor nodes powered by batteries from sleeping. By feeding incorrect inputs to the battery-powered computers, the attacker makes them alert. As a result, their batteries are exhausted, forcing them to shut down and causing network capacity to be disrupted [83].

(6)    Side-channel attack

A side-channel attack is a computer security attack that relies on knowledge obtained from the execution of a computer system rather than flaws in the algorithm itself. The attacker obtains the encryption keys using timing, control, and fault attacks on the system's computers [57].

(7)    Permanent denial of service attack (PDoS)

Denial of service through hardware sabotage is known as a permanent denial of service (PDoS) attack. Phishing is a general term for one way of launching a PDoS attack. An attacker damages a computer or hardware during such an attack, making the device or whole machine worthless [84].

*8.2. Network Layer Attacks*

(1)    Traffic analysis attacks

These attacks, such as eavesdropping attacks, rely on what the attacker hears in the network. Furthermore, without getting close to the network, the intruder will sniff confidential information or data flowing to and from the computers. However, the perpetrator must sacrifice the actual data in this attack [56].

(2) Wormhole attack

A wormhole attack is when an attacker creates a low-latency connection between two sensor nodes to mislead them, which impacts network routing, and then tunnels packets from one point to another across this link [85].

(3) RFID spoofing and unauthorized access

Because of the lack of proper authentication protocols, an attacker can read, alter, or erase data stored on RFID nodes. On the other hand, RFID spoofing entails reading and recording a data transmission from an RFID tag without being detected. When the data is re-sent, the original tag's Tag Identifier (TID) is used, making it appear accurate [86].

(4) Sybil attack

A peer-to-peer network attack is one in which a node executes several identities simultaneously, violating credibility schemes' legitimacy and control. In this case, a single malicious node takes on several roles (known as Sybil nodes) and transfers them around the network [87].

(5) Routing information attack

The routing information attack is a defunct network routing tactic to launch distributed denial of service (DDoS) reflection attacks against several targets, including direct attacks. The attacker spoofs or alters routing information and causes annoyance through events such as routing loops, error messages, and so on [88].

(6) Man-in-the-middle attack

The attack happens when an intruder introduces himself into a conversation between a receiver and an application, either to listen in or impersonate one of the participants. An attacker can listen in on or exploit an exchange between two IoT devices to access their private data while making it seem as if the conversation is expected [89].

(7) Selective forwarding

In this network attack, malicious nodes fail to facilitate data packets to prevent them from being transmitted to other nodes. All messages sent to other nodes in the network can be altered, dropped, or selectively forwarded by the malicious node. Consequently, the information sent to the intended receiver is inadequate [14].

(8) Replay attack

When a cybercriminal listens in on a protected network file, intercepts it, and fraudulently delays or re-sends it to the target, a DDoS assault results. A DoS assault is launched using a distributed strategy. [48].

(9) Blackhole attack

The attack in which all packets (control and data) routed through it are dropped by the malicious node. However, it is considered a DoS attack, but in conjunction with a blackhole attack and rank or sinkhole attacks, it becomes exceptionally hazardous [90].

(10) Denial/distributed denial of service attack

DDoS is a distributed denial of service (DDoS) attack launched by several sites simultaneously. A DDoS attack makes a web resource inaccessible to visitors by overloading the expected URL with more requests than the server can handle. Unlike a DoS attack, a DDoS attack requires multiple infected nodes to flood a single target with messages or connection requests to slow down or even crash the device server or network resource [91,92].

### 8.3. Software Attacks

(1)   Worms, viruses, Trojan horses

An adversary can infect a device with malicious software to tamper with data, steal information, or even conduct a denial-of-service attack. When you download a file or perform an update, a computer worm infects your computer. After that, the worm replicates and attacks other computers on the network. The Trojan horse infects your machine by downloading and opening a file. Unlike viruses, most Trojans are only present on your computer. When you email out files, a virus infects the host files on your machine and spreads to other users [93,94].

(2)   Malware

Malware is any software that may impair your computer equipment's output and functionality, either locally or remotely. Malware might contaminate the cloud or data centers if data from IoT devices were infected [95].

### 8.4. Data Attacks

(1)   Data breach

A data breach occurs when private, secure, or confidential information is stored, distributed, accessed, stolen, or used by someone who cannot do so. Memory leakage is the disclosure of personal, critical, or confidential information [17].

(2)   Unauthorized access

Unauthorized access occurs when a person uses another person's account or other methods to access a website, software, server, facility, or other equipment. Giving access to registered users and refusing entry to unauthorized users is what access management entails. Malicious users may obtain data control or privileged access through unauthorized access [96].

(3)   Data inconsistency

In the WSN-IoT, "data inconsistency" is a phrase used to describe an attack on data honesty that results in data inconsistency in transit or data kept in a central database [56].

## 9. WSN-IoT Security Mechanism in Industry 4.0

Numerous studies have been conducted on implementing effective security mechanisms in IR 4.0 applications. The research is mainly limited to research and development, which is why they lack real-time implementations; other limitations are security, performance, and testing. Most frameworks are based on academic research, so no real-time testing was conducted to evaluate their performance. Table 4 depicts the existing work related to security mechanisms and security solutions implemented by various researchers in WSN-IoT and maps these novel contributions with their limitations, which highlights the weak points of specific work to improve or overcome such limitations in the future or for the understanding of readers and researchers.

**Table 4.** WSN-IoT security mechanism solutions and their limitations.

| Author Name and Year | Proposed Solution | Limitations |
| --- | --- | --- |
| Geetanjali Rathee et al., 2020 [97] | Applying blockchain technology significantly increases wireless sensor security by comparing security metrics. It maintains worker confidentiality and accountability and tracks each sensor's operation. WSN-IoT artifacts and sensors are accumulated on the blockchain. The architecture is tested against the likelihood of an attack succeeding, the system's ability to detect an attack, and a falsification attack. | I time it takes to examine a single block before it is added to the blockchain is not mentioned. |

**Table 4.** *Cont.*

| Author Name and Year | Proposed Solution | Limitations |
| --- | --- | --- |
| Sahil Garg et al., 2019 [98] | The protocol's performance has been assessed, with the commonly used AVISPA, PUFs, ECC, and a host of other cryptographic primitives being used in the design of our stable, lightweight, and reliable authentication system for WSN-IoT environments, revealing that the proposed protocol supports shared authentication between WSN-IoT nodes and servers, as well as being resistant to a variety of security threats. | The protocIl is for research and academic purposes, and the implementation of the presented protocol has not been subjected to real-world application. |
| Jiafu Wan et al., 2016 [99] | Software-defined IIoT is proposed as a modern paradigm for industrial environments. It introduces features that make the network more scalable. The IIoT architecture can handle physical devices and an interface for information-sharing mechanisms via a conveniently customizable networking protocol. Additionally, the equipment's improved intelligence would boost the system's reliability and the availability of various services. | Information about the network attacks that the proposed software-defined IIoT can tackle is not mentioned. |
| Anichur Rahman et al., 2017 [100] | Industry 4.0 implementations in SDN-IoT-powered environments are faced with distributed blockchain-based authentication. The blockchain enhances the protection and privacy of Industry 4.0 services. The blockchain has the potential to lead in terms of security, secrecy, and confidentiality. Furthermore, the SDN-IoT integrates various Industry 4.0 services with increased security and stability. | The introduction of blockchain technology is also in its early stages. There is no study of the various threats, such as denial of service (DoS) and flooding attacks. Packet arrival rate, data response time, throughput, and other evaluation parameters are not specified. |
| Yulei Wu et al., 2020 [101] | Security and scalability have become top priorities in Industry 4.0, IoT, and IIoT. IoT and IIoT are implemented as vital technologies in Industry 4.0, and the blockchain and edge computing paradigms are briefly addressed. Integrating these two paradigms will contribute to developing stable and scalable critical infrastructure. The state-of-the-art WSN-IoT and IIoT infrastructure protection and privacy solutions and scalability solutions are evaluated and addressed. | Several possible research challenges and open problems are discussed but not adequately addressed or provided with a solution. |
| Jiafu Wan et al., 2018 [102] | Object Linking and Embedding (OLE) for process control technology, software-defined industrial networks, and device-to-device communications technology are combined with ontology modeling and multi-agent technology to achieve complex resource management. For innovative factory management for IoT-driven production, a load balancing mechanism based on Jena logic and Contract-Net Protocol technology provides a solution for complex resource utilization problems in Industry 4.0. | The sharing and reusing technologies of the ontology knowledge base are not mentioned without a scheduling system and algorithm optimization. |

## 10. Existing Literature

Many researchers have proposed their novel work in the domain of WSN-IoT. Table 5 shows the contributions of the recent work conducted by several researchers. The table states the contribution of their work and the limitations in terms of the research gaps of the specific research. The table identifies the research gaps, which helps spot the regions where further research and development must be conducted. The most significant research gaps are in the context of security, implementation, real-time processing, and evaluation. These gaps can provide future areas of work in this domain.

**Table 5.** Current work conducted in the domain of WSN-IoT and the research gaps.

| Sr.no | Author Name and Year | Contribution | Research Gaps |
|:---:|:---:|:---|:---|
| 1 | Susana H. Mellado et al., 2020 [103] | The technique has a low computing load and is very attractive for producing coverage maps that can be used for the optimum distribution of network resources. | The evaluation of output connectivity protocol requirements for satellite-based WSN-IoT simulations in virtual environments is lacking. |
| 2 | Trupti M. Behera et al., 2019 [91] | The algorithm considers residual energy, initial energy, and the cluster heads' desired value to select the following cluster heads for the network that fit WSN-IoT applications to maximize throughput, lifetime, and residual energy. | Does not consider further CH selection parameters in a mobile node network that continually changes its role. |
| 3 | Yifei Tan et al., 2019 [92] | Capable of collecting real-time data from an IoT-aided production device and creating a Design Thinking (DT) model that essentially represents the actual system's real situation. | Other essential features of DT are the lack of experimentation and optimization to forecast future positions or results. |
| 4 | Md. Ershadul Haque et al., 2020 [93] | The system is suitable for measurements of vibrations or earthquake events. The simulation results show that the EECDS-SGETRot mode offers the best products for a better version of the monitoring network and nodes that sinks can reach. | Not all of the nodes engage in data collection. |
| 5 | Zohre A. Bulaghi et al., 2020 [94] | The design reduces the amount of energy used by wireless sensors. The method is also helpful for determining the architectural location of sensors in WSNs, the total number of sensors required, and reliability. | The design contains some drawbacks and does not function on data in real-time. |
| 6 | Ravi Sharma & Shiva Prakash, 2020 [95] | Framework architecture for implementation patterns of relay nodes and load balancing between them to increase the network lifespan. This strengthens the deployment pattern of relay nodes for the sensor network. | The different network measurement criteria, such as network lifetime, overhead transmission costs, average electricity usage, and adequate overall area coverage, are still unknown. |
| 7 | Nabil D. et al., 2020 [90] | A new security mechanism, "Metric-based Routing Protocol for Low-power and Lossy Networks (RPL) Trustworthiness Scheme (MRTS)," is introduced to address the RPL vulnerabilities. Furthermore, the simulations indicate that MRTS is far more effective regarding energy consumption, packet delivery ratio, changes in node rank, and throughput. | The proposed strategies are highly complex since the reintegration of classified isolated nodes affects the model's efficacy and raises new issues. |

**Table 5.** *Cont.*

| Sr.no | Author Name and Year | Contribution | Research Gaps |
|---|---|---|---|
| 8 | S. Dasgupta and B. Saha, [104] | Machine learning was used to predict the types of DDoS attacks. XGBoost and Random Forest were used to reach the DDoS prediction system for the research. The UNWS-np-15 dataset from GitHub was simulated with Python. After using machine learning models to assess model performance, a confusion matrix was produced. The accuracy (PR) and recall (RE) of Random Forest in the first classification are 89%. The AC on this model is 89%, which is good. | It is imperative to provide a user-friendly, quicker substitute for deep learning calculations that delivers better results in a shorter amount of time. |
| 9 | Samy et al., [105] | Authors have slightly complicated the design of generic models to function successfully. The researcher has employed several different methods of categorization in their efforts to forecast DDOS assaults. Additionally, the authors used the support vector machine (SVM), the K-nearest neighbor (KNN) method, and the random forest (RF) algorithm. The SVM has a 99.5% accuracy rate for identifying DDoS assaults, whereas the KNN and RF have accuracy rates of 97.5% and 98.74%, respectively. | The author wants to work with real-time datasets, but this model only works with offline datasets. As a result, we must begin transferring this work to real-time fraud detection applications centered on supervised learning models. |

## 11. Comparative Analysis

Several approaches [44,50,57–59] have been developed in the literature related to routing mechanisms and protocols. Table 6 shows a comparative analysis, which shows that our proposed work discusses several factors, whereas others only discuss a few. Most existing works have not discussed anomaly types, challenges, advantages, and restraints regarding specific techniques. Information about attacks is also not addressed by most of the researchers. Finally, the total score has been calculated to measure the highest factor rate for computing purposes. However, this work is novel in terms of the factors mentioned earlier, as other existing research has not been as comprehensive work, especially in terms of IoT-LPN research challenges, LPN architecture, and LPN application.

**Table 6.** Comparative analysis with existing research.

| Factors | Dragomir [7] | Gendreau and Moorman [14] | Iqbal et al. [45] | Dalipi and Yayilgan [50] | Alansari et al. [88] | Anbar, et al. [106] | Adefemi Alimi et al. [107] | Raza et al. [108] | Muzammal et al. [109] | This Paper |
|---|---|---|---|---|---|---|---|---|---|---|
| LPN architecture | 0 | 0 | 0 | 0 | 0 | 0 | 1 | 0 | 1 | 1 |
| LPN application | 0 | 0 | 0 | 0 | 1 | 0 | 1 | 1 | 1 | 1 |
| Physical layer attacks | 0 | 0 | 0 | 0 | 1 | 0 | 0 | 0 | 0 | 1 |
| Network layer attacks | 0 | 1 | 1 | 0 | 1 | 1 | 1 | 1 | 1 | 1 |
| Software attacks | 0 | 0 | 0 | 0 | 0 | 0 | 1 | 0 | 0 | 1 |
| Data attacks | 0 | 0 | 0 | 0 | 0 | 0 | 0 | 0 | 0 | 1 |
| WSN issues and challenges | 0 | 1 | 0 | 0 | 1 | 0 | 0 | 1 | 1 | 1 |
| IoT-LPN research challenges | 0 | 0 | 0 | 0 | 0 | 0 | 1 | 1 | 0 | 1 |
| Security objectives/factor | 1 | 0 | 1 | 1 | 0 | 1 | 1 | 1 | 1 | 1 |

**Table 6.** *Cont.*

| Factors | Dragomir [7] | Gendreau and Moorman [14] | Iqbal et al. [45] | Dalipi and Yayilgan [50] | Alansari et al. [88] | Anbar, et al. [106] | Adefemi Alimi et al. [107] | Raza et al. [108] | Muzammal et al. [109] | This Paper |
|---|---|---|---|---|---|---|---|---|---|---|
| Protocols discussion | 1 | 0 | 0 | 1 | 0 | 1 | 1 | 1 | 0 | 1 |
| Industry 4.0 | 0 | 0 | 0 | 0 | 0 | 0 | 0 | 0 | 0 | 1 |
| Total Score | 2 | 2 | 2 | 2 | 4 | 3 | 7 | 6 | 5 | 11 |

## 12. Conclusions and Findings

The WSN-IoT is recognized as a technology that has already had economic consequences and generated the hope that different sectors' competitive domains will change dramatically. In the manufacturing industry, sensor-equipped machines can collect data from the production system in real-time and use it to store and synchronize real-world data on the cyber side. It is possible to continually develop and synchronize with the natural world by changing data in real-time and contrasting cyberspace with physical space simultaneously. IIoT, which acts as an inevitable IR 4.0 infrastructure, imposes high reliability, low latency, scalability, energy consumption, and security requirements. However, many low-power wireless specifications and infrastructure protocols aim to meet and conform to those criteria. The emphasis is on addressing the current problems and pursuing future exploration avenues to discover potential solutions. In particular, the "smart" side of IR 4.0 is focused on the availability of self-awareness, self-management, and self-healing intelligent networks. That includes technical concerns such as real-time cellular networking, 5G cellular networks, shallow power usage, and industrial cyber-security. The most trending WSN-IoT security mechanism techniques are blockchain, cryptography algorithms, and AI techniques. These can be an excellent solution to this problem and provide accuracy, autonomy, security, and efficient results.

This article aims to provide a detailed overview of WSN-IoT's characteristics and aspects for low-powered IoT mechanisms; the paper throws light on the research gaps, including WSN-IoT issues and challenges. According to our findings, researchers have proposed WSN-IoT security mechanisms. However, the significant limitations are that these solutions are still under study and in progress. As previously stated, most of the work conducted so far is in its early stages, is only for research or academic purposes, and does not involve working in real-time scenarios. Considerable research has been for educational purposes and lacks proper deployment on a large scale and testing in the natural environment. Furthermore, frameworks' functional and large-scale deployment are significant limitations in the context of current research and contributions.

## 13. Future Work

The current low-powered WSN-IoT protocols also have various limitations. Further, the work so far lacks implementation strictness and real-time security techniques or measurement parameters, which can help identify the security level and provide an analysis. These limitations and research gaps have created a whole new space and opportunity for researchers to develop new solutions. New efficient security mechanisms and testing techniques are needed to ensure these security techniques' integrity and quality. Furthermore, in the future, this also requires improvements on the work already undertaken as there is a requirement for real-time and large-scale systems.

**Author Contributions:** Formal analysis, M.Z.H.; Investigation, M.Z.H.; Methodology, M.Z.H.; Visualization, M.Z.H.; Writing—original draft, M.Z.H.; Writing—review & editing, M.Z.H. Project administration, Z.M.H.; Funding acquisition, Z.M.H.; Supervision, Z.M.H. All authors have read and agreed to the published version of the manuscript.

**Funding:** This work was supported by Geran Putra Berimpak Universiti Putra Malaysia, Vote Number 9659400. Our sincere thanks to Geran Putra Berimpak Universiti Putra Malaysia for their support.

**Data Availability Statement:** Not applicable.

**Conflicts of Interest:** The authors declare no conflict of interest.

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
