# Peer review of "Efficient Secure Routing Mechanisms for the Low-Powered IoT Network: A Literature Review"

_electronics, doi:10.3390/electronics12030482_

Round 1

Reviewer 1 Report

The article is well drafted and good contribution to the field of Low Powered IoT Network. However, there are some issues that need attention of the author to improve quality of the manuscript.

1.     The paper has includes many URL links in the references. As the current study is a literature review, so It is suggested to change them to some good journal citations.

2.     Redraw all diagrams and visualize ranking of the categories so as the readers clearly understand each category supported by how many research articles.

3.     There are some grammatical mistakes in the submission. Please use Grammarly to cross-verify all mistakes.

4.     Add machine learning-related research in your Literature review & how this can help you get good results.

5.       Try to add some more comparison on the already work done in this domain.

6.       It will be better to add more relevant references to support the categories.

Author Response

  1. The paper has includes many URL links in the references. As the current study is a literature review, so It is suggested to change them to some good journal citations.

Reply : Changes incorporated

  1. Redraw all diagrams and visualize ranking of the categories so as the readers clearly understand each category supported by how many research articles.

Reply : Changes incorporated

  1. There are some grammatical mistakes in the submission. Please use Grammarly to cross-verify all mistakes.

Reply : Changes incorporated

  1. 4.Add machine learning-related research in your Literature review & how this can help you get good results.

Reply : Changes incorporated

  1. 5.Try to add some more comparison on the already work done in this domain.

Reply : Changes incorporated

  1. It will be better to add more relevant references to support the categories.

Reply : Changes incorporated

Reviewer 2 Report

The present paper surveys recent trends in the efficient routing mechanisms for low-powered IoT networks. To that end, it compiles a detailed list of literature references and compares modern algorithms.

Strengths

Survey papers are generally a good fit in fast-growing fields with many facets, such as IoT routing protocols. The authors analyze the application of such routing algorithms in real-world scenarios and identify that differential evolution is one of the key technologies. 

Weaknesses

While I understand that the paper aims to review the current literature and, thus, does not claim to present novel ideas, I still think that the paper's contribution is rather small. In particular, the paper describes many things outside the title of the paper. The title especially describes the efficient routing protocols for IoT; however, the rest of the paper talk about the security concerns in IoT. Further, sometimes the author talks about the IoT- LPN, and RPL, which makes it difficult for the reader to understand the article's main point.  

The contributions of this work need to be clarified. I went through the abstract and introduction, but I did not get the main contribution of this work. The authors should spend significant efforts to enhance the main work in this research. 

Title: There is no link between the title and the main text as it reviews the security issues in IoT.

Abstract: The authors should improve the abstract. The abstract should contain the problem, background, review methodology, and results. In addition, the abstract is very long and has some language errors.

Introduction: misleading, storytelling is messing.

The introduction, e.g., should lead the way throughout the paper. In addition, the benefits coming from this research should be made clearer in the introduction and throughout the paper. This section needs significant efforts to make things and contributions clearer and the contents flow.

The last paragraph in the introduction should summarize the paper.

Enhance the figure quality. 

Authors should include statistical information about the articles related to the proposed study (Number of papers, publishers, journals, conferences, chapters, etc.).

What is the link between the low-power routing protocols and the security concerns in IoT?

Sections 3 and 4 have the same title.   

Section 6 There are no references to support your claims.

The language has some typos, which should be improved to ensure the readers' readability.

Author Response

Abstract: The authors should improve the abstract. The abstract should contain the problem, background, review methodology, and results. In addition, the abstract is very long and has some language errors.

Introduction: misleading, storytelling is messing.

The introduction, e.g., should lead the way throughout the paper. In addition, the benefits coming from this research should be made clearer in the introduction and throughout the paper. This section needs significant efforts to make things and contributions clearer and the contents flow.

The last paragraph in the introduction should summarize the paper.

Enhance the figure quality. 

Authors should include statistical information about the articles related to the proposed study (Number of papers, publishers, journals, conferences, chapters, etc.).

What is the link between the low-power routing protocols and the security concerns in IoT?

Sections 3 and 4 have the same title.   

Section 6 There are no references to support your claims.

There are number of references

The language has some typos, which should be improved to ensure the readers' readability.

Reviewer 3 Report

Overall, paper is written well but there are some concerns which need to be addressed:

1. Change the title of the paper

2. Summarize abstract 

3. Add some more lines about the contribution points in the introduction

4. Add the below titled paper in the section IoT in Industry 

Smart IoT Control-Based Nature Inspired Energy Efficient Routing Protocol for Flying Ad Hoc Networks 

5. Explain more table 4

6. Explain figure 2

7. Separately write Future directions 

Author Response

Overall, paper is written well but there are some concerns which need to be addressed:

  1. 1. Change the title of the paper

Reply : Changes incorporated

  1. Summarize abstract 

Reply : Changes incorporated

  1. Add some more lines about the contribution points in the introduction

Reply : Changes incorporated

  1. Add the below titled paper in the section IoT in Industry 

Smart IoT Control-Based Nature Inspired Energy Efficient Routing Protocol for Flying Ad Hoc Networks 

  1. Explain more table 4

Reply : Changes incorporated

  1. Explain figure 2

Reply : Changes incorporated

  1. Separately write Future directions 

Round 2

Reviewer 2 Report

- Authors don't respond to reviewer comments.
- The list of author responses is copying and pasting the reviewer's comments without a response.

Author Response

Reviewer 2

The present paper surveys recent trends in the efficient routing mechanisms for low-powered IoT networks. To that end, it compiles a detailed list of literature references and compares modern algorithms.

Strengths

Survey papers are generally a good fit in fast-growing fields with many facets, such as IoT routing protocols. The authors analyse the application of such routing algorithms in real-world scenarios and identify that differential evolution is one of the key technologies. 

Weaknesses

While I understand that the paper aims to review the current literature and, thus, does not claim to present novel ideas, I still think that the paper's contribution is rather small. In particular, the paper describes many things outside the title of the paper. The title especially describes the efficient routing protocols for IoT; however, the rest of the paper talk about the security concerns in IoT. Further, sometimes the author talks about the IoT- LPN, and RPL, which makes it difficult for the reader to understand the article's main point.  

The contributions of this work need to be clarified. I went through the abstract and introduction, but I did not get the main contribution of this work. The authors should spend significant efforts to enhance the main work in this research. 

Title: There is no link between the title and the main text as it reviews the security issues in IoT.

Reply: The title is already appropriate and approved by my committee.

Abstract: The authors should improve the abstract. The abstract should contain the problem, background, review methodology, and results. In addition, the abstract is very long and has some language errors.

Reply: Revised as recommended by the reviewer

Introduction: misleading, storytelling is messing

The introduction, e.g., should lead the way throughout the paper. In addition, the benefits coming from this research should be made clearer in the introduction and throughout the paper. This section needs significant efforts to make things and contributions clearer and the contents flow.

The last paragraph in the introduction should summarize the paper.

Enhance the figure quality. 

Authors should include statistical information about the articles related to the proposed study (Number of papers, publishers, journals, conferences, chapters, etc.).

Reply:  Changed as recommended by the reviewer in the revised article.

What is the link between the low-power routing protocols and the security concerns in IoT?

Reply:   The low-power routing protocols used in Internet of Things (IoT) devices are designed to minimize the power consumption of the devices, which is important for devices that are powered by batteries or have limited access to power. However, this focus on power efficiency can also create security concerns.

One concern is that low-power routing protocols may not have the same level of security as more power-hungry protocols. For example, they may not include encryption or authentication features, or they may use weaker forms of encryption. This can make it easier for attackers to intercept and modify communication between IoT devices.

Another concern is that the use of low-power routing protocols may make it more difficult to update the security of IoT devices. Because these devices are designed to minimize power consumption, they may not have the resources or connectivity necessary to receive regular security updates. This can leave them vulnerable to attacks that exploit known vulnerabilities.

Overall, it is important for IoT device manufacturers and users to carefully consider the trade-offs between power efficiency and security when selecting low-power routing protocols, and to take steps to secure their devices and networks. This can include implementing strong encryption and authentication, regularly updating the devices with security patches, and following best practices for securing IoT networks.

Sections 3 and 4 have the same title.   

Reply:  Changes incorporated

Section 6 There are no references to support your claims.

Reply: There are number of references.

The language has some typos, which should be improved to ensure the readers' readability.

Reply:  Changes incorporated.

Round 3

Reviewer 2 Report

After reviewing the revised version, I recommend the authors consider the following comments to enhance the paper:
1) Identify the meaning of the following terms:
- WSN-IoT and RPL in the abstract.
- CPS in line 166 instead of line 181.
- IoE line 220.
2) Re-write the paragraph from lines 128 to 138, especially lines 128 to 131.
3) Summary of the paper structure at the end of the introduction.
4) Lines from 171 to 196, please, the references of the claims. Similarly, lines 200 to 219.
5) Line 222, remove "." after the word Similarly.
 6) I recommend the authors discuss briefly or summarize the finding of table 1. Or no need to section 5, as it's discussed in table 1.
7) Avoid using bulk references, for example, line 280.
8) In section 8, modify the items number, for example, 2. ) Malicious Code injection attack, use the "." or ")", not both.
9) Enhance table 8 and explain the findings of the table clearly.

Author Response

After reviewing the revised version, I recommend the authors consider the following comments to enhance the paper:
1) Identify the meaning of the following terms:
- WSN-IoT and RPL in the abstract.
- CPS in line 166 instead of line 181.
- IoE line 220.

Reply: Changes incorporated

2) Re-write the paragraph from lines 128 to 138, especially lines 128 to 131.

Reply: Changes incorporated

3) Summary of the paper structure at the end of the introduction.

Reply: Changes incorporated

4) Lines from 171 to 196, please, the references of the claims. Similarly, lines 200 to 219.

Reply: Changes incorporated

5) Line 222, remove "." after the word  Similarly.

Reply: Changes incorporated

 6) I recommend the authors discuss briefly or summarize the finding of table 1. Or no need to section 5, as it's discussed in table 1.

Section 1 is about Effective Low Powered and LLN protocols in IoT Frameworks, section 5 is about Categories of network attacks in WSN-IOT layers. One is talking about the frame works and other is about categories of attacks.
7) Avoid using bulk references, for example, line 280.

It’s a comparison that’s why I used bulk references,
8) In section 8, modify the items number, for example, 2. ) Malicious Code injection attack, use the "." or ")", not both.

Could not understand this one
9) Enhance table 8 and explain the findings of the table clearly.

Reply: Changes incorporated

Round 4

Reviewer 2 Report

Accept in present form.